# Evaluation of Ruthenium(II) *N*-Heterocyclic Carbene Complexes as Enzymatic Inhibitory Agents with Antioxidant, Antimicrobial, Antiparasitical and Antiproliferative Activity

**DOI:** 10.3390/molecules28031359

**Published:** 2023-01-31

**Authors:** Ibrahim S. Al Nasr, Waleed S. Koko, Tariq A. Khan, Nevin Gürbüz, Ismail Özdemir, Naceur Hamdi

**Affiliations:** 1Department of Biology, College of Science and Arts, Qassim University, Unaizah 51911, Saudi Arabia; 2Department of Science Laboratories, College of Science and Arts, Qassim University, Ar Rass 51921, Saudi Arabia; 3Department of Clinical Nutrition, College of Applied Health Sciences, Qassim University, Ar Rass 51921, Saudi Arabia; 4Department of Chemistry, Faculty of Science and Art, İnönü University, Malatya 44280, Turkey; 5Catalysis Research and Application Center, İnönü University, Malatya 44280, Turkey; 6Department of Chemistry, College of Science and Arts at ArRass, Qassim University, Ar Rass 51921, Saudi Arabia

**Keywords:** N-Heterocyclic carbene, 5,6-dimethylbenzimidazolium salts, ruthenium, HCT-116, HepG-2, antimicrobial, antioxidant, enzyme inhibition, *Leishmania*, *Toxoplasma*

## Abstract

A series of [RuCl_2_(p-cymene)(NHC)] complexes were obtained by reacting [RuCl_2_(p-cymene)]2 with in situ generated Ag-N-heterocyclic carbene (NHC) complexes. The structure of the obtained complexes was determined by the appropriate spectroscopy and elemental analysis. In addition, we evaluated the biological activities of these compounds as antienzymatic, antioxidant, antibacterial, anticancer, and antiparasitic agents. The results revealed that complexes **3b** and **3d** were the most potent inhibitors against AchE with IC_50_ values of 2.52 and 5.06 μM mL^−1^. Additionally, **3d** proved very good antimicrobial activity against all examined microorganisms with IZ (inhibition zone) over 25 mm and MIC (minimum inhibitory concentration) < 4 µM. Additionally, the ligand **2a** and its corresponding ruthenium (II) complex **3a** had good cytotoxic activity against both cancer cells HCT-116 and HepG-2, with IC_50_ values of (7.76 and 11.76) and (4.12 and 9.21) μM mL^−1^, respectively. Evaluation of the antiparasitic activity of these complexes against *Leishmania major* promastigotes and *Toxoplasma gondii* showed that ruthenium complexes were more potent than the free ligand, with an IC_50_ values less than 1.5 μM mL^−1^. However, **3d** was found the best one with SI (selectivity index) values greater than 5 so it seems to be the best candidate for antileishmanial drug discovery program, and much future research are recommended for mode of action and in vivo evaluation. In general, Ru-NHC complexes are the most effective against *L. major* promastigotes.

## 1. Introduction

Currently, the use of organometallic and inorganic compounds is very common in contemporary medication [1,2]. New organometallic complexes called N-Heterocyclic Carbene (NHC) complexes show promise drug formulation [3,4,5,6,7]. Çetinkaya et al. revealed the results of the first study on the biological functions of NHC complexes [8,9,10,11]. For this reason, several research teams have synthesized functionalized NHC complexes and investigated their biological activities [12,13,14,15,16,17]. In this regard, complexes of the ruthenium (II/III) type have been thoroughly studied as DNA binding, antibacterial, and anticancer agents [18,19,20,21]. In particular, ruthenium-type compounds have been investigated against various cancer cell lines as prospective substitutes for the well-known diamine-dichloroplatinum (II) in the formulation of novel anticancer medicines (cisplatin) [22]. Under physiological circumstances, ruthenium can access the +2 and +3 oxidation states and can bind to cells’ proteins, nucleic acids, sulfur, or oxygen-containing molecules [23,24,25,26,27,28,29]. Additionally, depending on the characteristics of their ligand, ruthenium complexes can optimize the kinetics of their interactions with cell components. As a result, ruthenium complexes ligands exchange rates are near to those of biological processes, making them ideally suited for use in a variety of therapeutic contexts. Therefore, ruthenium compounds may have greater cytostatic activity than platinum-based medications against a variety of cancer cells and may also be helpful against cisplatin-resistant cancer cells. In addition, a brand-new class of highly intriguing physiologically active compounds known as ruthenium complexes has developed. Among the most dangerous human pathogens, *Staphylococcus aureus* is an eminent human pathogen that can colonize the human host and cause severe life-threatening illnesses [30,31,32,33,34,35,36,37].

Our research group was also investigating metal complexes with anticancer activity against a various types of cancer cell lines [38,39,40]. We are currently investigating new functional NHC ligands that provide a favorable environment for the development and utilization of metal compounds. In this paper, we synthesized and characterized a new series of Ru(I) NHC complexes containing benzimidazoles. The structure of the new compound was characterized by various spectroscopic and analytical methods. Next, enzyme inhibition against AChE and TyrE, antioxidant against 2,2-diphenyl-1-picrylhydrazyl (DPPH), 2,2’-azino-bis(3-ethylbenzothiazoline-6-sulfonic acid) (ABTS) and β-carotene bleaching test, various biological activities such as antimicrobial activity against Gram-positive, Gram-negative and *Candida albicans*, antiproliferative activity against colon cancer cell lines (HCT-116) and hepatocellular carcinoma cell lines (HepG-2), as well as antiparasitic activity against *Leishmania major* and *Toxoplasma gondii*, and cytotoxicity against Vero cells, were examined.

## 2. Results

### 2.1. Chemistry

#### 2.1.1. Preparation of Benzimidazolium Salts **2a**–**d**

The synthesis of benzimidazolium salts (**2a**–**d**) as NHC precursors was carried out as previously described [41] (Figure 1). By using ^1^H NMR, ^13^C NMR, FT-IR, and elemental analysis, the structures of the benzimidazolium salts **2a**–**d** were confirmed.

The ^1^H NMR spectra of precursors **2a**–**d** show characteristic downfield shifts in the range δ 9.85–11.83 ppm for the NCHN protons due to the positive charge of the molecules [42].

The assigned structure was further supported by the benzimidazolium salt’s ^1^H NMR spectra. Sharp singlets representing the C(2)-H resonances were detected at 9.85, 10.34, 11.83, and 11.58 ppm for **2a**–**c**, respectively. Chemical shifts measured by ^13^C NMR agreed with the suggested structure. At 141.3, 143.0, 143.1, and 152.8 ppm, the imino carbon is a characteristic singlet for the 1H decoupling mode of the benzimidazolium bromides **2a**–**d**. The aliphatic area of the ^13^C NMR spectra displayed a sequence of peaks in the range of 20.76–35.18 ppm corresponding to resonances of the aliphatic carbon nucleus, whereas aromatic rings were seen in the range of 113.38–152.82 ppm. These numbers are fairly consistent with data that have already been published [43,44].

#### 2.1.2. Preparation of Ruthenium-Carbene Complexes **3a**–**3d**

By transmetallating the corresponding silverNHC derivatives without isolation, the novel [RuCl_2_(p-cymene)(NHC)] complexes (**3a**–**3d**) were prepared using a two-step procedure. By then adding [RuCl_2_(p-cymene)]2 to the mixture, orange–brown complexes were obtained with high yields (80–90%). In contrast to nonpolar solvents, chloroform, dichloromethane, tetrahydrofuran, and ruthenium carbene complexes (**3a**–**3d**) are soluble in these solvents. Figure 2 provides the synthesis and the structures of Ru(II)-NHCs complexes. Using spectroscopic methods such as ^1^H NMR, ^13^C NMR, and IR as well as elemental studies, the structures of complexes **3a**–**3d** were determined.

The aromatic protons of complexes **3a**–**3d** appeared as multiplets between 5.57–5.88 and 7.13–7.69 ppm, the methyl protons appeared between 0.92–1.16 and 2.27–2.61 ppm as singlets. In all complexes (**3a**–**3d**), the -CH proton of the p-cymene group was seen as a heptet in the 2.64–2.86 ppm range. (NCH_2_) exhibited a doublet resonance in the ^1^ H NMR spectra of (**3a**–**3d**) between 4.21 and 5.01 and 5.29 and 5.31 ppm. The carbene carbon in the ruthenium complexes **3a**–**3d** exhibits 13C chemical shifts at 189.2, 187.8, 188.9, and 189.0 ppm, respectively. The values obtained are comparable to those that have been published for other Ru-NHC complexes [45,46]. The ruthenium complexes **3a**–**3d** were also validated by elemental analysis results.

### 2.2. Biological Evaluation

#### 2.2.1. Enzymatic Inhibitory, AChE and TyrE Inhibitory Activity

The results shown in Table 1 indicate that complexes **3b** and **3d** were the most potent inhibitors against AchE with IC_50_ values of 2.52 and 5.06 μM mL^−1^.

#### 2.2.2. Antioxidant Activity

A significant antioxidant activity, comparable to that of the conventional BHT, was found for compound **3d**. For the DPPH, ABTS, and -carotene tests, the IC_50_ values of this compound were 32.18, 18.17, and 92.25 µM mL^−1^, respectively (Table 2). For the DPPH, ABTS, and -carotene tests, the standard BHT’s IC_50_ values were 31.55, 17.41, and 89.55 µM mL^−1^, respectively (Table 2).

#### 2.2.3. Antimicrobial Activity

Figure 1 displays the antibacterial activity of the corresponding ruthenium (II) complexes (**3a**–**3d**) and the synthetic NHC ligands (**2a**–**2d**). All of the examined indicator organisms were inhibited by complexes **3b** and **3d**. Inhibition zones caused by complex **3d** against *S. aureus*, *L. monocytogenes*, *E. coli*, *P. aeruginosa*, *S. typhimurium*, and *C. albicans* are 26, 28, 29, 27, 27, and 27 mm, respectively (Figure 1).

The MIC ranges from 1.95 to 62.5 µM mL^−1^ for *S. aureus* and *S. typhimurium*, from 3.9 to 62.5 µM mL^−1^ for *L. monocytogenes*, and from 1.25 to 31.25 µM mL^−1^ for *C. albicans*. Complexes **3b** and **3d** gave the lowest MIC values. The MIC values for synthetic compound **3b** against *L. monocytogenes*, *S. aureus*, *S. typhimurium*, and *C. albicans* were 15.6, 3.9, 3.9, and 1.25 µM mL^−1^, respectively (Figure 2). In relation to complex **3d**, the MIC values against *L. monocytogenes*, *S. aureus*, *S. typhimurium*, and *C. albicans* were 3.9, 1.95, 1.95, and 1.25 µM mL^−1^, respectively (Figure 2). Additionally, Figure 2 shows that complex **3d**’s MIC value against *L. monocytogenes* was the same as that of conventional ampicillin (3.9 g).

#### 2.2.4. Antiproliferative Activity

Screening of the selected compounds against human colon carcinoma cancer cell lines and hepatocellular carcinoma cell lines revealed that the compound ruthenium(II) complex **3a** had IC_50_ values (4.12 and 9.21 µM mL^−1^) in both human cancer cell lines where the mentioned values were approximately equivalent to those of standard vinblastine drugs (3.83 and 6.05 µM mL^−1^) in cytotoxic activity Table 3.

#### 2.2.5. Antiparasitical Activity

##### Antileishmanial Results

From Table 4, we can observe that all the compounds revealed antileishmanial activity against both the amastigote and promastigote stages. For amastigotes, all the compounds gave IC_50_ values less than 4.3 µM mL^−1^, and only Compound **2b** showed IC_50_ values less than 1 µM mL^−1^ (0.3 µM mL^−1^). All ruthenium(II) complexes (**3a**–**d**) gave IC_50_ values less than 1 µM mL^−1^ against *L. major* promastigotes. However, complexes **3c** and **3d** are the most active against *L. major* promastigotes, with SI values over five. There are strong similarities for the cytoxicity results of all compounds, with CC_50_ values in the range of 1.1 to 2.9 µM mL^−1^. Therefore, only two compounds, **3a** and **3d**, can be recommended for future use as antileishmanial agents.

##### Antitoxoplasmal Results

Table 5 indicates that 4 compounds (ruthenium (II) complexes **3a**–**d**) showed antitoxoplasmic activity with IC_50_ values ≤ 1.5 µM mL^−1^, which were 1.3, 1.4, 1.5 and 1.4 µM mL^−1^, respectively. However, their SI was less than 1.5, which indicates their toxicity for Vero cells that can limit their future uses for drug formulation.

## 3. Materials and Methods

All procedures were carried out under an inert atmosphere using standard Schlenk line techniques according to our previous work [38,39,40].

### 3.1. Synthesis of Ligands (**2a**–**d**)

A mixture of benzimidazolium salt 1 (1 g) and the corresponding benzyl bromide (1eq) in DMF (2 mL) was stirred at 70 °C for 2–3 days. After that time, the white solid formed was washed with diethyl ether (20 mL) and stirred for couple hours. Then, the reaction mixture was filtred through filter paper, and the white solid was dried under vacuum, then crystallized with DCM-ether (1:3) for further purification.


**5,6-dimethyl-1,3-bis(2,3,4,5,6-pentamethylbenzyl)-1H-benzo[d]imidazol-3-ium bromide 2a**


m.p. 307.7 °C. Yield (96%). ν(CN) = 1440.99 cm^−1^. ^1^H NMR (CDCl_3_, 400MHz) δ (ppm) 2.22(s, 30H, CH_3_), 2.28(s, 6H, CH_3_(a,b)), 5.8(s, 4H, CH_2_(_1′,1″_)), 7.11(s, 2H, Harom(4,7)),9.85(s, 1H, H_2_). ^13^CNMR(CDCl_3_, 101MHz) δ (ppm) 17.01(CH_3_), 16.92(C_(c,g,c’,g’)_), 17.01 (C(_d,f,d’,f’_)), 17.28(C(e,e’)), 20.77(C_(a,b)_), 48.10_(C1′;1″)_, 113.41_(C4;7)_, 125.58_(C8;9)_, 130.42(C_4′;5′;6′;4″;5″;6″_), 133.50(C_3′;7′;3″;7″_), 133.74(C_5;6_), 136.88(C_2′;2″),_141.35(C_2_). Anal. Calcd for C_31_H_39_BrN_2_:C, 71.66%; H, 7.57%; N, 5.39%%. Found: C, 71.17; H, 7.8; N, 5.4%.


**5,6-dimethyl-1,3-bis(2,4,6-trimethylbenzyl)-1H-benzo[d]imidazol-3-ium bromide 2b**


m.p. 210.6 °C. Yield (92%). ν(CN) = 1454.84 cm^−1^. ^1^H NMR (CDCl_3_, 400MHz) δ (ppm) 2.15(s, 12H, CH_3_(a,e,a’,e’)), 2.18(s, 12H, CH_3_(b,d,b’,d’)), 2.20(s, 6H, CH_3_(c,c′)), 5.84(s, 4H, CH_2_), 7.2(d, 2H, Harom(5,6)), 7.34(d, 2H, Harom(4,7)), 10.34(s, 1H, H_2_). ^13^C NMR(CDCl_3_,100MHz) δ (ppm) 16.95(C_a,e,a’,e’_), 17.06(C_b,d,b’,d’_), 17.30(C_c,c’_), 48.58(C_1′;1″_), 113.72(C_4;7_), 125.36(C_5;6_),126.88(C_8;9_),131.85(C_5′;5″_),133.49(C_4′;6′;4″;6″_), 133.82(C_3′;7′;3″;7″_), 137.07(C_2′;2″_)_,_ 143(C_2_). Anal. Calcd for C_29_H_33_BrN_2_:C, 71.16%; H, 6.80%; N, 5.72%. Found: C, 71.3; H, 6.9; N, 5.8%.


**5,6-dimethyl-1,3-((4-(tert-butyl)-4-methyl-benzyl)-1H-benzo[d]imidazol-3-ium bromide 2c**


m.p. 296.5 °C. Yield (89%). ν(CN) = 1427.88 cm^−1^. ^1^H NMR (CDCl_3_, 400MHz) δ (ppm) 1.2 (s, 18H, CH_3_), 5.75(s, 4H, CH_2_), 7.33(d, 4H, Harom(4′,6′,4″,6″)),7.39(d, 4H, Harom(3′;7′3″,7″)), 7.46(d, 2H, Harom(5;6)), 7.55(d, 2H, Harom(4;7)), 11.83(s, 1H, H_2_). ^13^CNMR(CDCl_3_,101 MHz) δ (ppm) 31.18(CH_3_), 34.67(C_8′;8”_), 51.27(C_1′;1”_), 113.78(C_4;7_), 126.34(C_4′;6′;4″;6″_), 127.09(C_5,6_), 128.17(C_3′;7′;3″;7″_), 129.56(C_8;9_),131.41(C_2′;2″_), 143(C_2_),152.46(C_5′;5″_). Anal. Calcd for C_28_H_33_BrN_2_:C, 70.43%; H, 6.80%; N, 5.72%. Found: C, 70.2; H, 6.9; N, 5.8%.


**5,6-dimethyl-1,3-(3,5)-dimethyl-4-methylbenzyl)-2,3-dihydro-1H-benzo[d]imidazolium bromide (2d)**


Yield: 92%; M.p. = 235 °C; FT-IR (KBr) ν(CN)(cm^−1^): = 1565 (C=N); 1359 (C-N) cm^−1^ ; ^1^H NMR (CDCl_3_,300 MHz) δ (ppm):= 11.58 (s, 1H, H_2_, NCHN); 7.44 (d, 2H, H_4″, 6″,_ arom. CH, 3 JHH = 7.43 Hz); 7.37 (s, 1H, H_5′_, arom. CH); 7.35 (d, 2H, H_3″, 7″,_ arom. CH, 3 JHH = 7.34 Hz); 7.27 (s, 1H, H_3′_, arom. CH); 7.01 (s, 2H, H_4,7_, arom. CH); 6.94 (s, 1H, H_7′_, arom. CH); 5.76 (s, 2H, H_1′_,CH_2_); 5.67 (s, 2H, H_1″_,CH_2_); 2.26 (s, 6H, H_d,e_, 2 × CH_3_); 1.25 (s, 9H, Ha,b,c, 3×CH_3_). ^13^C NMR (CDCl_3_, 75MHz) (δ (ppm)):152. 81 (C_2_, NCN); 142.37 (C_4′, 6′_, arom.Cq); 139.59 (C_2′_, arom.Cq); 137.84 (C_5″_, arom. CH); 133.11 (C_2″_, arom.Cq); 131.35 (C_5, 6_, arom.Cq); 130.45 (C_8, 9_, arom.Cq); 128.53 (C_4″, 6″_, arom. CH); 126.77 (C_3′, 5′, 7′_, arom. CH); 126.23 (C_3″, 7″_, arom. CH); 113.82 (C_4, 7_, arom. CH); 51.80 (C_1′_, CH_2_); 51.43 (C_1″_, CH_2_); 35.17 (C_d, e_, 2 × CH_3_); 31.70 (Cc, CH_3_); 21.75 (C_a, b_, 2 × CH_3_); Anal. Calcd for C_27_H_31_BrN_2_:C, 69.97%; H, 6.74%; N, 6.04%. Found: C, 70.1; H, 6.8; N, 6.1%

Synthesis of ruthenium N-heterocyclic Carbene complexes (**3a**–**3d**): The synthesis was performed according to our previous work [38].


**5,6-Dimethyl-[1,3-(2,3,4,5,6-pentamethyl-2,4,6-trimethyl)-benzimidazol-2-ylidene](p-cymene) ruthenium(II) chloride, (3a)**


Yield: (80%); m.p. = 204 °C. FT-IR(KBr)ν(CN)(cm^−1^) = 1404 (C-N).^1^HNMR (300 MHz, CDCl_3_, δ(ppm)): 7.13 (d, 2H, p-CH_3_C_6_H_4_CH(CH_3_)_2_); 7.00 (d, 2H, p-CH_3_C_6_H_4_CH(CH_3_)_2_); 6.92 (s, 1H, CH_2_C_6_H_3_(CH_3_)2-3,5); 6.84 (d, 4H, CH_2_C_6_H_4_(CH_3_)-4); 6.67 (s, 2H, CH_2_C_6_H_3_(CH_3_)2-3,5); 6.51 (m, 2H, CH_2_C_6_H_4_(CH_3_)—4); 5.57 (m, 2H, C_6_H_2_(CH_3_)2-5,6); 5.30 (s, 2H, H_1′_, CH_2_); 5.02 (s, 2H, H_1″_, CH_2_); 2.64 (p, 1H, H _7‴_pCH_3_C_6_H_4_CH(CH_3_)_2_); 2.34 (s, 3H, He, CH_3_); 2.30 (s, 6H, Hc, d, 2 × CH_3_); 2.19 (s, 6H, Ha, b, 2 × CH_3_); 1.82 (s, 3H, Hf, CH_3_); 1.58 (s, 3H, Hi, CH_3_); 1.15 (s, 6H, Hg, h, 2 × CH_3_ (p-CH_3_C_6_H_4_CH(CH_3_)_2_).^13^C NMR (CDCl_3_, 75 MHz) (δ (ppm)): 189.2 (C_2_, NCN); 138.4; 138.2; 137.0; 135.0; 134.5; 134.3; 132.3; 129.6; 129.0; 125.8; 123.6; 111.9; 107.4; 96.6; 85.2; 52.4 (C_1′_, CH_2_); 52.2 (C_1″_, CH_2_); 30.6 (C_7‴_, pCH_3_C_6_H_4_CH(CH_3_)_2_); 21.6 (C_h, g_, 2 × CH_3_, p-CH_3_C_6_H_4_CH(CH_3_)_2_); 21.2 (C_c, d_, 2 × CH_3_); 22.3 (C_e, f_, 2 × CH_3_); 18.1 (C_a, b, i_, 3 × CH_3_). Anal. Calcd for C_41_H_53_RuN_2_Cl_2_:C, 66.02%; H, 7.16%; N, 3.76%. Found: C, 66.1; H, 7.3; N, 3.8%.


**5,6-Dimethyl-[1,3-(2,3,4,5,6-pentamethyl-2,4,6-trimethyl)-benzimidazol-2-ylidene](p-cymene) ruthenium(II) chloride, (3b)**


Yield: (88%); m.p. = 184 °C. FT-IR(KBr)ν(CN)(cm^−1^) = 1418 (C-N).^1^HNMR(300 MHz, CDCl_3_, δ(ppm)): 7.69 (d, 2H, p-CH_3_C_6_H_4_CH(CH_3_)_2_); 7.54 (d, 2H, p-CH_3_C_6_H_4_CH(CH_3_)_2_); 7.26 (s, 1H, CH_2_C_6_H_3_(CH_3_)_2_-3,5); 6.89 (s, 2H, CH_2_C_6_H_3_(CH_3_)2); 6.63 (s, 2H, C_6_H_2_(CH_3_)2); 5.88 (s, 1H, CH_2_C_6_H(CH_3_)4); 4.21 (s, 4H, H_1′,1″_, 2 × CH_2_); 2.86 (p, 1H, H _7‴_, p-CH_3_C_6_H_4_CH(CH_3_)_2_); 2.27 (s, 18H, H_a, b, c, d_, f, g, 6 × CH_3_); 2.20 (s, 3H, Hi, CH_3_); 1.95 (s, 3H, H_l_, CH_3_); 1.32 (s, 6H, He,h, 2 × CH_3_); 0.92 (s, 6H, Hj, k, 2 × CH_3_ (p-CH_3_C_6_H_4_CH(CH_3_)_2_). ^13^C NMR (CDCl_3_, 75 MHz) (δ (ppm)): 187.8 (C_2_, NCN); 138.4; 135.4; 135.2; 134.5; 133.5; 133.2; 132.3; 131.6; 131.3; 130.0; 128.9; 124.6; 123.4; 107.1; 96.2; 54.0 (C1′, CH_2_); 51.8 (C_1”_, CH_2_); 31.0 (C_7‴_, p-CH_3_C_6_H_4_CH(CH_3_)_2_); 21.5 (C_j, k_, 2 × CH_3_, p CH_3_C_6_H_4_CH(CH_3_)_2_); 20.8 (C_c, d_, 2 × CH_3_); 20.2 (Ci, CH_3_); 18.4 (C_f, g_, 2 × CH_3_); 16.3 (C_a, b_, 2 × CH_3_); 15.4 (C_e, h, l_, 3 × CH_3_). Anal. Calcd for C_41_H_53_RuN_2_Cl_2_:C, 66.02%; H, 7.16%; N, 3.76%. Found: C, 66.1; H, 7.3; N, 3.8%.


**5,6-Dimethyl-[1,3-(4-(tert-butyl)-4-methyl)-benzimidazol-2-ylidene] (p-cymene) ruthenium (II) chloride, (3c)**


Yield: (87%); m.p. = 224 °C. FT-IR(KBr)ν(CN)(cm^−1^) = 1609 (C-N). ^1^HNMR(300 MHz, CDCl_3_, δ(ppm)): 7.36 (d, 4H, CH_2_C_6_H_4_C(CH_3_)3-4); 7.03 (d, 4H, CH_2_C_6_H_4_C(CH_3_)3-4); 6.82 (s, 2H, C_6_H_2_(CH_3_)2-5,6); 6.55 (d, 2H, p CH_3_C_6_H_4_CH(CH_3_)_2_); 5.62 (d, 2H, p- CH_3_C_6_H_4_CH(CH_3_)_2_); 5.31 (s, 2H, H _1′_, CH_2_); 5.02 (s, 2H, H _1″_, CH_2_); 2.61(s, 6H, H_d, g_, 2 × CH_3_); 2.19(s, 6H, H_a, b_, 2 × CH_3_); 1.81 (s, 3H, Hi, CH_3_); 1.66 (s, 1H, H _7‴_, p-CH_3_C_6_H_4_CH(CH_3_)_2_); 1.31 (s, 15H, H_c, e, f, h, l_, 5 × CH_3_); 1.12 (d, 6H, H_j, k_, 2 × CH_3_ (pCH_3_C_6_H_4_CH(CH_3_)_2_). ^13^CNMR (CDCl_3_, 75 MHz) (δ (ppm)): 188.9 (C_2_, NCN); 150.4; 134.9; 134.4; 132.3; 125.8; 125.5; 111.9; 106.8; 97.3; 85.5; 52.3 (C_1′,1″,_ 2 × CH_2_); 41.1 (C7‴, p-CH3C6H4CH(CH3)_2_); 34.6 (Cd, g, 2 × CH_3_, CH_2_C_6_H_4_C(CH_3_)_3_); 31.4 (C_c, e, f,h_, 4 × CH_3_); 30.5 (Ci, CH_3_); 22.6 (C_a, b_, 2 × CH_3_); 20.3 (C_j, k_, 2 × CH_3_). DART-TOF-MS: *m*/*z* = 453, 426, 291. Anal. Calcd for C_38_H_46_RuN_2_Cl_2_:C, 64.95%; H, 6.60%; N, 3.99%. Found: C, 65.1; H, 6.7; N, 4.1%.


**5,6-Dimethyl-[1,3-(3,5)dimethyl-4-methyl)-benzimidazol-2-ylidene](p-cymene) ruthenium(II) chloride, (3d)**


Yield: (90%); m.p. = 210 °C. FT-IR(KBr)ν(CN)(cm^−1^)= 1409 (C-N). ^1^HNMR(300 MHz, CDCl_3_, δ(ppm)): 14 (d, 4H, CH_2_C_6_H_4_(CH_3_)-4); 7.00 (d, 4H, CH_2_C_6_H_4_(CH_3_)); 6.79 (s, 2H, C_6_H_2_(CH_3_)2); 6.46 (d, 2H, p-CH_3_C_6_H_4_CH(CH_3_)_2_); 5.68 (d, 2H, p-CH_3_C_6_H_4_CH(CH_3_)_2_); 5.29 (s, 2H, H _1′_, CH_2_); 5.03 (s, 2H, H_1″_, CH_2_); 2.68 (p, 1H, H _7‴_, p-CH_3_C_6_H_4_CH(CH_3_)_2_); 2.34 (s, 6H, H_c, d_, 2 × CH3); 2.18 (s, 6H, Ha, b, 2 × CH_3_); 1.84 (s, 3H, He, CH_3_); 1.16 (d, 6H, H_f, g_, 2 × CH_3_ (p-CH_3_C_6_H_4_CH(CH_3_)_2_). ^13^CNMR (CDCl_3_, 75 MHz) (δ (ppm)): 189.0 (C_2_, NCN); 137.0; 134.9; 134.4; 132.3; 129.6; 125.9; 111.9; 107.6; 97.0; 85.30; 52.5 (C_1′,1″_, 2 × CH_2_); 30.6 (C_7‴_, p-CH_3_C_6_H_4_CH(CH_3_)_2_); 22.6 (C_f, g_, 2 × CH_3_); 21.2 (C_c, d_, 2 × CH_3_); 20.3 (Ce, CH_3_); 18.3 (C_a, b_, 2 × CH_3_); 14.2 (CH, CH_3_).

### 3.2. Biological Activities

#### 3.2.1. Enzymatic Inhibitory Assay

##### Acetylcholinesterase Inhibitory (AChEI)

According to the Ellman et al. published spectrophotometric method of electric eel AChE [47], AChEI activity was measured. Acetylthiocholine iodide (ATCI) was used as the reaction’s substrate, and the antiacetylcholinesterase activity was measured using 5,5’-Dithiobis-(2-nitrobenzoic acid) (DTNB).

##### Antityrosinase Activity

According to Rangkadilok et al. [48], the TyrE inhibitory activity was measured spectrophotometrically using L-tyrosine as the substrate in a 96-well microplate.

#### 3.2.2. Antioxidant Activity

Three methods were used for the assessing antioxidant properties of the selected compounds, which are 2.2-diphenyl-1-picrylhydrazyl (DPPH), 2.2’-azino-bis (3-ethylbenzothiazoline-6-sulphonic acid) (ABTS) radicals scavenging, and β-carotene linoleic acid bleaching assay. Antioxidant activity was expressed as IC_50_ (the concentration that causes 50% of inhbition effect). The control compound was butylated hydroxytoluene (BHT), which is a potent antioxidant.

##### DPPH Radical Scavenging Activity

Briefly, various amounts of produced compounds were diluted with ultrapure water after being dissolved in dimethylsulfoxide (DMSO)/water (1/9; *v*/*v*) (1, 0.5, 0.250, 0.125, 0.0625, 0.03125 mg mL^−1^). The samples were then combined with 500 mL of a 4% (*w*/*v*) solution of the DPPH radical in ethanol. The combination was incubated for 30 min. at room temperature and in the dark [49]. Spectrophotometric analysis was used to calculate the scavenging capacity by comparing the decrease in absorbance at 517 nm to a blank.

##### ABTS Assay

ABTS radical scavenging activity was conducted by referring to the method of Re et al. [50].

##### β-Carotene Bleaching Assay

β-Carotene bleaching test was conducted following the method described by Pratt’s [51].

All the assays used for antioxidant determination (DPPH, ABTS, and β-carotene bleaching assay) were performed simultaneously three times in the same conditions. The results obtained in µM mL^−1^ average of the three experiments.

#### 3.2.3. Antimicrobial Activity

Microorganisms, media and growth conditions, agar well diffusion method for inhibition zone determination (IZ) and minimum inhibitory concentration (MIC) were performed according to the literature work [52,53].

The synthesized compounds were examined in vitro for their antimicrobial activity against Six standard microorganisms of ATCC, two Gram positive bacteria *S. aureus* and *Listeria monocytogenes*, three Gram negative bacteria *Esherichia coli, Pseudomonas aeruginosa* and *Salmonella typhimurium* and the fungus *C. albicans.* Bacteria were cultured in Luria-Bertani (LB) medium, while Sabouraud agar was used for culturing *C. albicans* and the assay conducted according to our previous techniques [53]. The results were the average of 3 readings.

#### 3.2.4. In Vitro Anticancer Proliferation Studies

The selected compound was investigated for its cytotoxic properties against HCT-116 and HepG-2 (cancer cell lines of ATCC, Rockville, MD, USA). Vinblastine was applied as reference drug. The assay was conducted according to the methods described by Mossman [54] and our recently published data [55].

The results presented IC_50_ (The concentration that causes 50% inhibitory of cell viability) of µM mL^−1^ from the average of 3 reading.

#### 3.2.5. Antiparasitical Assessment

##### Leishmania Major Cell Isolation, Culture Conditions, and Assays

This assay was carried out according to the methods mentioned in our previously published article [56]. *L. major* promastigotes were isolated locally from an indoor patient in 2016, liquid nitrogen was used for the preservation of the parasites, and BALB/c mice were used for the maintenance of the parasites and production of *L. major* amastigotes. Phenol red-free RPMI 1640 medium (Invitrogen, USA) with 10% FBS was used for the culture and in vitro evaluation, while amphotericin B (AmB) was used as reference drug. The result was expressed in IC_50_ values (the concentration that causes 50% inhibition of the viable parasites) of three independent readings, followed by the selectivity index (SI) calculation by dividing CC_50_ (toxic concentration that causes 50% inhibition of cell growth) over IC_50_ of the same compound [56].

##### Toxoplasma Gondii Cell Line, Culture Conditions, and Assay

This assay was carried out according to the methods mentioned in our previously published article [50]. Vero cells line (ATCC^®^ CCL81™, USA) were used for the serial passage and cultivation of *T. gondii* tachyzoites RH strain, complete RPMI 1640 medium with heat-inactivated 10% FBS was used for the culture and in vitro evaluation, while atovaquone (ATO) was used as reference drug. The results were expressed in IC_50_ of three independent readings, followed by the selectivity index (SI) calculation by dividing CC_50_ over IC_50_ of the same compound [56].

#### 3.2.6. In Vitro Cytotoxicity Assay

MTT colorimetric technique was carried out for cytotoxicity evaluation according to the methods mentioned in our previously published article [57]. An amount of 96 well plates with complete were used for the culture of the cells. FLUOstar OPTIMA spectrophotometer was applied for colorimetric analysis and in vitro evaluation. Cytotoxic effects were expressed by CC_50_ values (concentration that caused a 50% reduction in viable cells), from three independent experiments [56].

## 4. Conclusions

In summary, ruthenium(II)-NHC complexes **3a**–**3d** have been easily prepared by the reaction of silver(I)- NHC complexes as a carbene transfer reagent with [RuCl2(p-cymene)]2 in dichloromethane at room temperature in good yields. The molecular structures of the benzimidazolium salts (**2a**–**2d**) and the Ru(II)–N-heterocyclic carbene (NHC) complexes **3a**–**3d** were characterized by elemental analysis and 1H- and ^13^C-NMR spectra.

The results of the enzymatic inhibitory study against AChE and TyrE revealed that complexes **3b** and **3d** are the most effective inhibitors against AchE, with respective IC_50_ values of 2.52 and 5.06 µM mL^−1^ and 19.88 and 24.95 µM mL^−1^. These results confirm that NHC metallic complexes have potent antibacterial properties [58]. Important antioxidant activity was observed for Complex **3**. The synthesized NHC ligands (**2a**–**2d**) and their corresponding ruthenium(II) complexes (**3a**–**3d**) were screened against HCT-116 and HepG-2, and the results revealed that ruthenium(II) complex **3a** exhibited cytotoxic activity approximately equivalent to that of standard vinblastine, so we can suggest ruthenium(II) complex **3a** can be used in the formulation of drugs that stimulate cancer treatment against human colon carcinoma cancer and liver hepatocellular carcinoma cancer after further pharmacological and clinical trials investigations. Regarding the last experiment of studying the ruthenium (II) complex as an antiparasitical agent against *L. major* and *T. gondii*, compounds **3c** and **3d** were found to have extremely potent antileishmania effects, with a SI over five, while all tested compounds had less antitoxoplasmic activity. These findings were similar to our previous investigation with NHC palladium complexes as well as the similar ruthenium complexes [53,57]. We propose that **3d** can be used as a drug candidate for many antimicrobial, anticancer, and antiparasite bioactivities, and further investigation for mode of action detection and in vivo evaluation is highly recommended.

## Data Availability

The datasets generated during and/or analyzed during the current study are available from the corresponding author on reasonable request.

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
