# Peer review of "Evaluation of Ruthenium(II) N-Heterocyclic Carbene Complexes as Enzymatic Inhibitory Agents with Antioxidant, Antimicrobial, Antiparasitical and Antiproliferative Activity"

_molecules, 2023, doi:10.3390/molecules28031359_

Round 1
Reviewer 1 Report
The results are clearly presented however the techniques for determining the MIC for bacteria lack completely. This part should include : where the strains come from, the medium used, the condition of growth, (temperature, incubation time).
The diameters must have been measured on agar plate (to be detailed) but how MIC values were determined: if on agar plates which reference has been used to rely inhibition diameter to product concentration. If done in liquid medium, what were the concentrations of the product solutions used ? Are the dilutions the same that the ones used for parasites. Indicate also the number of replicates, this not only for the samples but also for the antibiotics used for comparison. Why antibiotics are not present in the figure 1. In the figure 2 what is the yellow square here for?
How authors explain the great discrepancies between the indicated inhibition diameters and MIC results. The discussion should be a little developed. Is the proposed compound have an advantage when compared to other metallic compounds and if yes, why.
Third minor notices. What was the mM concentration of the solution of the different compounds, Were they adjusted at the same level? Indicate the meaning of the abbreviation SI. It is unusual to have the Material an Methods part not included in the text. At least an ouline may be given...
All the names of bacteria have also to be written in italics in full at their first apparition in the text and abbreviated afterwards (e.g. Escherichia coli and E.coli) .
Author Response
AUTHORS’ RESPONSES TO THE REVIEWERS’ COMMENTS
Manuscript ID: molecules-2135206
Title:
Evaluation of Ruthenium(II) N-Heterocyclic Carbene Com-plexes as Enzymatic Inhibitory Agents with Antioxidant, An-timicrobial, Antiparasitical and Antiproliferative Activity
Journal:
Molecules
Dear editor,
Thank you very much for the comments, recommendations, and corrections
We have carefully checked the following items suggested by the editor, and we have re-arranged the main text according to the suggestions. All revisions have been given with the Yellow color.
We sincerely hope that you will find our revised manuscript suitable for publication as an origin article in your prestigious journal.
Looking forward to receiving a prompt reply,
Sincerely,
Pr. Naceur HAMDI
Corresponding author
Reply to the reviewers’ comments
|
Reviewer Number |
Original comments of the reviewer |
Reply by the author(s) |
|
#1 |
The results are clearly presented however the techniques for determining the MIC for bacteria lack completely. This part should include : where the strains come from, the medium used, the condition of growth, (temperature, incubation time).
|
An outline for the methods was added to the text according to your valuable comments and observation, all the details are presented or may be the reference mentioned contain all the details you asked about. Between page 2 – 7 |
|
The diameters must have been measured on agar plate (to be detailed) but how MIC values were determined: if on agar plates which reference has been used to rely inhibition diameter to product concentration. If done in liquid medium, what were the concentrations of the product solutions used ? Are the dilutions the same that the ones used for parasites. Indicate also the number of replicates, this not only for the samples but also for the antibiotics used for comparison.
|
Details were found in the methods section of the text. The results were taken from 3 replicates for all the experiments. |
|
|
Why antibiotics are not present in the figure 1. In the figure 2 what is the yellow square here for? |
Antibiotics with well-known results against all used strains, so we used them only in the MIC The yellow square is omitted |
|
|
How authors explain the great discrepancies between the indicated inhibition diameters and MIC results. The discussion should be a little developed. Is the proposed compound have an advantage when compared to other metallic compounds and if yes, why.
|
When we take the IZ results 3d and 3b gave the higher IZ diameter (over 25 mm), the same results were found when MIC was used for evaluation 3d and 3b gave the lowest values for the MIC (that indicate no discrepancies present in the results). only 3 bacteria were for MIC evaluation as supporting indicators for the values of IZ. |
|
|
Third minor notices. What was the mM concentration of the solution of the different compounds, Were they adjusted at the same level? Indicate the meaning of the abbreviation SI. It is unusual to have the Material an Methods part not included in the text. At least an ouline may be given...
|
Actually we use serial concentration of compounds, when we put outlines of methods we give the references used in our previously published data. |
|
|
All the names of bacteria have also to be written in italics in full at their first apparition in the text and abbreviated afterwards (e.g. Escherichia coli and E.coli) .
|
Was done for all microorganisms and parasites |
|
|
|
|
|
|
#2 |
1. In the manuscript, the author combines the result with the discussion, and it is recommended that the author separate the discussion from the result.
|
Was done. The discussion is separated from results in a specific section |
|
2. The title of the article should be more precise and the word "evaluation" was inappropriate
|
The title is approved from the committee of Scientific research at Qassim University, so we cannot change it. |
|
|
3. L44, L58: the reference should be placed before the full stop, please check the full text.
|
Was done |
|
|
4. Materials is not found in the article and data analysis method, please add them in detail.
|
An outline for the methods was added to the text according to your valuable comments and observation |
|
|
5. L11-112: What substance is evaluated by biological evaluation? How is it evaluated?
|
We mean the selected compounds |
|
|
6. To find out how the IC50 values are measured, what is the significance of their values? |
All these were mentioned in the section of the methods, that added to the articles |
|
|
7. The callout should be placed below the picture. |
Done for all figures |
|
|
8. The article pointed out that 3b and 3d were the most effective inhibitors of AChE, and why the conclusion only suggested further research on 3d.
|
Due to its activity against other microorganisms, cancer cells and parasites. Also it is less toxic against vero cells 1.8 while 3b 1.3 uM. |
|
|
9. Line 164-166: ruthenium(II) complex 3a can induce apoptosis. Is there any literature or data supporting this? |
The sentence shifted from results to discussion and recommendation part. |
|
|
10.The overall discussion of the article is not deep enough and the layout is confusing. 2 is the result and discussion, 3-5 is only the result, and the author is suggested to further adjust. |
Each activity was discussed in separate but some compounds sharing many good activities, that is why we recommend 3d in the last. |
|
|
|
|
|
|
#3 |
The biological functions reported here seem to be interesting. However, the authors show only results without discussion in the text, and this manuscript is simply regarded as a technical report, not a scientific paper. The authors are requested to show their research concept or molecular design and dictate the substituent effects of the carbene ligands in detail in the discussion part.
|
An outline of methods was added in a separate part, also the discussion is separated from the results according to the valuable comments of all review. We recommend in the future research for mode of action detection to elaborate the exact activity reasons. |
|
|
In addition, the authors have published several papers on the similar compounds. The authors should show the relevance of the following papers and the present paper at least. L. Boubakri, A. Chakchouk-Mtiba, O. Naouali, L. Mellouli, L. Mansour, I. Özdemir, et al.: Ruthenium(II) complexes bearing benzimidazole-based N-heterocyclic carbene (NHC) ligands as potential antimicrobial, antioxidant, enzyme inhibition, and antiproliferative agents. J. Coordin. Chem. 75, 2022 - Issue 5-6 .
|
Was done, in the methods and discussion Actually these papers were prepared in the same time, and this paper published later after this article was written. Thank you for this valuable observation. |
|
|
Abstract: Many symbols and abbreviated names are used here. They must be shown in full names. There are many abbreviations that are not well defined and making the manuscript less understandable. For example: P. 2, line no. 65: “acetylcholinesterase (AChE) and tyrosinase (TyrE)”
|
Was done In the methods the used enzymes were explained |
|
|
P. 11, The references numbered [57] and [58] are absent |
Were added |

Reviewer 2 Report
The ruthenium (II) N-heterocyclic carbene complexes as enzymatic inhibitory agents with antioxidant, antimicrobial, antiparasitical and antiproliferative activity were evaluated in this manuscript, which provided a reference for investigating their biological activities. There are some issues in this manuscript that need to be fixed.
1. In the manuscript, the author combines the result with the discussion, and it is recommended that the author separate the discussion from the result.
2. The title of the article should be more precise and the word "evaluation" was inappropriate
3. L44, L58: the reference should be placed before the full stop, please check the full text.
4. Materials is not found in the article and data analysis method, please add them in detail.
5. L11-112: What substance is evaluated by biological evaluation? How is it evaluated?
6. To find out how the IC50 values are measured, what is the significance of their values?
7. The callout should be placed below the picture.
8. The article pointed out that 3b and 3d were the most effective inhibitors of AChE, and why the conclusion only suggested further research on 3d.
9. Line 164-166: ruthenium(II) complex 3a can induce apoptosis. Is there any literature or data supporting this?
10.The overall discussion of the article is not deep enough and the layout is confusing. 2 is the result and discussion, 3-5 is only the result, and the author is suggested to further adjust.
Author Response

(The authors gave the same response as above.)

Reviewer 3 Report
This manuscript deals with the biological activities of [RuCl2(p-cymene)(NHC)] complexes in which a series of N-heterocyclic carbene (NHC) are coordinated with Ru. These complexes were evaluated as anti-enzymatic, antioxidant, antibacterial, anticancer and antiparasitic agents. The biological functions reported here seem to be interesting. However, the authors show only results without discussion in the text, and this manuscript is simply regarded as a technical report, not a scientific paper. The authors are requested to show their research concept or molecular design and dictate the substituent effects of the carbene ligands in detail in the discussion part.
In addition, the authors have published several papers on the similar compounds. The authors should show the relevance of the following papers and the present paper at least.
L. Boubakri, A. Chakchouk-Mtiba, O. Naouali, L. Mellouli, L. Mansour, I. Özdemir, et al.: Ruthenium(II) complexes bearing benzimidazole-based N-heterocyclic carbene (NHC) ligands as potential antimicrobial, antioxidant, enzyme inhibition, and antiproliferative agents. J. Coordin. Chem. 75, 2022 - Issue 5-6
Abstract: Many symbols and abbreviated names are used here. They must be shown in full names.
There are many abbreviations that are not well defined and making the manuscript less understandable. For example:
P. 2, line no. 65: “acetylcholinesterase (AChE) and tyrosinase (TyrE)”
P. 11, The references numbered [57] and [58] are absent.
Author Response

(The authors gave the same response as above.)

Round 2
Reviewer 1 Report
[47] H. Jelali, W. Koko, S. M. Al-Hazmy, L. Mansour, J. Al-Tamimi, E. Deniau,... & N.Hamdi, Journal of Chemistry, 2022, 548 2022.. This reference has to be completed
Many other references lack tittle. Thus, all the references are to be checked and completed if necessary according to editor recommendations for the authors.
Otherwise, the remarks have been taken into account, thus the paper is now acceptable once references have been corrected .
Author Response
Dear Professor,
thank you for your comments, all the references were checked and the titles were added. I hope that this version will be ok
best
NH
Reviewer 3 Report
The authors have revised their manuscript by considering the reviewers' comments.
Author Response
Dear Professor,
all the titles of the papers were added.
Best regards
NH